# Probiotics Support Resilience of the Oral Microbiota during Resolution after Experimental Gingivitis—A Randomized, Double-Blinded, Placebo-Controlled Trial

**DOI:** 10.3390/nu15224805

**Published:** 2023-11-16

**Authors:** Christine Lundtorp Olsen, Laura Massarenti, Vincent Frederik Dahl Vendius, Ulvi Kahraman Gürsoy, Annina Van Splunter, Floris J. Bikker, Mervi Gürsoy, Christian Damgaard, Merete Markvart, Daniel Belstrøm

**Affiliations:** 1Section for Clinical Oral Microbiology, Department of Odontology, Faculty of Health and Medical Sciences, University of Copenhagen, 2200 Copenhagen, Denmark; vincent.vendius@sund.ku.dk (V.F.D.V.); mema@sund.ku.dk (M.M.); dbel@sund.ku.dk (D.B.); 2AMD Denmark A/S, 3390 Hundested, Denmark; 3Section for Oral Biology and Immunopathology, Department of Odontology, Faculty of Health and Medical Sciences, University of Copenhagen, 2200 Copenhagen, Denmark; laura.massarenti@sund.ku.dk (L.M.); chrd@sund.ku.dk (C.D.); 4Department of Periodontology, Institute of Dentistry, University of Turku, 20520 Turku, Finland; ulvi.gursoy@utu.fi (U.K.G.); mervi.gursoy@utu.fi (M.G.); 5Department of Oral Biochemistry, Academic Center for Dentistry Amsterdam, University of Amsterdam and VU University Amsterdam, 1081 LA Amsterdam, The Netherlands; a.p.van.splunter@acta.nl (A.V.S.); f.bikker@acta.nl (F.J.B.)

**Keywords:** dental plaque, gingivitis, microbiota, probiotics, randomized controlled trial, 16s ribosomal RNA

## Abstract

The present study aims to test whether probiotics protect against experimental gingivitis incited by 14 days of oral hygiene neglect and/or subsequently support the restoration of oral homeostasis. Eighty systemically and orally healthy participants refrained from oral hygiene procedures for 14 days, followed by 14 days with regular oral hygiene procedures. Additionally, participants consumed either probiotics (*n* = 40) or placebo (*n* = 40) throughout the trial. At baseline, day 14, and day 28, supragingival plaque score and bleeding-on-probing percentage (BOP %) were registered, and supragingival plaque and saliva samples were collected. The supragingival microbiota was characterized using 16S sequencing, and saliva samples were analyzed for levels of pro-inflammatory cytokines and proteases. At day 28, the relative abundance of *Lautropia* (*p* = 0.014), *Prevotella* (*p* = 0.046), *Fusobacterium* (*p* = 0.033), and *Selenomonas* (*p* = 0.0078) genera were significantly higher in the placebo group compared to the probiotics group, while the relative abundance of *Rothia* (*p* = 0.047) species was associated with the probiotics group. *Streptococcus sanguinis* was associated with the probiotics group, while *Campylobacter gracilis* was associated with the placebo group. No difference was observed in salivary cytokines, albumin, or any enzyme activity. The present study suggests that probiotics support the resilience of the oral microbiota in the resolution period after gingivitis.

## 1. Introduction

Biofilm-induced gingivitis, here referred to as gingivitis, is among the most prevalent diseases worldwide, affecting up to 90% of the population [1]. While gingivitis is reversible, it is also the precursor of periodontitis. In most cases, adequate self-performed oral hygiene is sufficient for the prevention of gingivitis [1,2]. Yet, gingivitis remains a highly prevalent disease, which is why additional approaches such as probiotic supplements have been suggested as a method of prevention [3,4,5,6].

Since 1965, the experimental gingivitis model has been extensively used for studies on the pathogenesis of the disease [7], and recently, the experimental gingivitis model has also been employed to test the potential protective effect of the supplementary use of probiotics [3,4,5,6]. In general, the data point toward a protective effect of probiotic supplements, but the limited number of studies together with the use of different probiotic strains makes the results rather inconclusive at this stage. Importantly, previous studies have focused on clinical, microbiological, or immunological data without combining them [3,4,5,6], which complicates the possibility of characterizing the full effect of tested probiotics on the oral microbiota and host response. Considering that gingivitis is a multifactorial disease that is reflected in clinical, microbiological, and immunological endpoints, there is a need for probiotic studies to characterize the effect on all three parameters.

Recently, we demonstrated that the consumption of probiotic lozenges containing *Lacticaseibacillus rhamnosus* PB01 DSM14870 and *Lactilactobacillus curvatus* EB10 DSM32307 combined with xylitol was able to induce statistically significant health-associated compositional changes to the salivary microbiota in orally healthy individuals, which was antagonistic to sugar-mediated dysbiosis [8]. Yet, it remains to be tested whether this probiotic supplement has a protective effect during disease-associated perturbations to the oral cavity, such as biofilm-induced gingivitis.

In a randomized, double-blinded, clinical trial, we, therefore, tested the hypothesis that supplementary use of probiotics protects the supragingival microbiota in healthy individuals against biofilm-induced gingivitis caused by a 14-day absence of oral hygiene procedures and that probiotics support the resilience of the supragingival microbiota in the resolution period, i.e., the 14 days after oral hygiene procedures are resumed. Specifically, we aimed to test whether the combination of *L. rhamnosus* PB01 DSM14870 and *L. curvatus* EB10 DSM32307 combined with xylitol protects the oral cavity against the microbiological consequences of a 14-day absence of oral hygiene procedures and/or supports the restoration of oral homeostasis during the resolution period. Additionally, we aimed to test whether probiotics have a clinical impact as measured according to the levels of plaque and bleeding on probing (BOP %). Further, we looked at the ability of probiotics to regulate the microbiological composition through immune response activation as measured according to the levels of the selected cytokines and proteases. Hence, the present study evaluates the impact on clinical, microbiological, and immunological parameters.

## 2. Materials and Methods

### 2.1. Study Design

The present study is a quadruple-blinded (Participant, Care Provider, Investigator, Outcomes Assessor), randomized, placebo-controlled trial conducted from 5 January to 25 May 2022, at the Department of Odontology, University of Copenhagen. The trial had a total duration of 28 days, and through computerized randomization (www.randomizer.org, accessed on 3 January 2022) DB randomly allocated participants at baseline to receive either probiotics or placebo lozenges twice a day throughout the trial period. According to randomization, identical pots containing either probiotic or placebo lozenges were numbered 1–80. The randomization code was blinded to participants as well as the examinator (CLO). All participants refrained from all oral-hygiene procedures during the initial 14 days (±2 days) of the trial period, after which oral hygiene procedures were resumed in the subsequent 14 days (±2 days) (Figure 1). Clinical examinations and collection of samples were performed at baseline, on day 14, and on day 28 (±2 days). The trial was performed according to the Helsinki Declaration, registered at ClinicalTrials.gov (UCPH_01_006), and reported to the local data authorization of the Faculty of Health and Medical Sciences, University of Copenhagen (514-0434/19-3000). This study was approved by the regional ethical committee (H-21003295). All participants signed informed consent before participation.

### 2.2. Study Population

The study population consisted of 80 systemically and orally healthy individuals aged 19–33 years. The sample size was based on post hoc analyses from a previous experimental gingivitis study [9] showing that a sample size of *n* = 20 was sufficient to detect a significant impact of oral hygiene neglect on the supragingival microbiota. As we speculated the potential protective effect of probiotics on the microbiota to be 50%, a sample size estimate revealed that *n* = 20 + (20 × 0.5) = 30 were needed in each group. To ensure sufficient power despite dropouts of as much as 30%, *n* = 40 were enrolled in each group. Participants were recruited at the Department of Odontology, University of Copenhagen, and continuously received a number from 1 to 80. The inclusion criteria were systemically and orally healthy adults aged between 18 and 35 years. Exclusion criteria were active dental caries or periodontitis, extensive gingivitis (>20%), pregnancy, systemic diseases, use of medication, and systemic antibiotics within the last three months.

### 2.3. Collection of Samples

Paraffin-stimulated saliva samples and supragingival plaque samples were collected at baseline, day 14, and day 28 (±2 days) on weekdays between 8 a.m. and 5 p.m. as previously described with minor modifications [8,9,10,11,12]. In brief, participants refrained from consuming any food or drink for two hours before the appointment to avoid stimulation of the salivary glands before the collection of the paraffin-stimulated saliva samples. Secondly, supragingival plaque samples were collected from the buccal surface of the second quadrant. To secure enough sample material participants avoided oral hygiene procedures in the morning on trial days. Samples were immediately stored at −18 °C and within 8 h removed to −80 °C.

### 2.4. Clinical Data

Clinical examinations were performed by the same examinator (CLO) in continuation of sample collection at baseline, day 14, and day 28 (±2 days). Levels of plaque were measured as previously described [10,11] using SUNSTAR G·U·M^®MD^ RED-COTE^®MD^ disclosing tablets and graded from 0 to 5 using the Modified Quigley and Hein index [13]. Bleeding on probing was categorized from 0 to 2, and bleeding percentage was calculated as previously reported [10,11]. At baseline and day 28, participants performed their normal oral hygiene procedure (brushing and interdental cleaning twice a day) after the collection of supragingival plaque samples and before clinical examinations. On day 14, plaque accumulation during oral hygiene neglect was registered before oral hygiene procedures were performed.

### 2.5. Probiotic and Placebo Lozenges

The probiotic and placebo lozenges along with the instructions were the same as previously used and described aside from the addition of xylitol [8,11]. In brief, the probiotic lozenges contained an equal mix of *L. rhamnosus* PB01 DSM14870 and *L. curvatus* EB10 DSM32307 with a concentration of 1 × 10^9^ CFU/tablet and 491 mg xylitol. Participants were instructed verbally and in writing to soak and distribute one lozenge right after toothbrushing and avoid drinking or food consumption for the following 30 min. The lozenges were packed in identical pots, and the examiner as well as the participants were blinded throughout the trial.

### 2.6. DNA Extraction, Library Preparation, and DNA Sequencing

DNA extraction and sequencing library preparation were performed as previously described [8,11]. In brief, the 16S gene was targeted in the V1–V3 region using MiSeq (Illumina, San Diego, CA, USA). Samples yielding significantly less-quality filtered DNA reads (filtReads) than 10,000, here, filtReads < 8000, failed and were disregarded in all subsequent analyses. Bioinformatic processing of microbiological data was matched against the 16S rRNA Human Oral Microbiome RefSeq database (HOMD) v 15.2 [14] as previously described [8,10,11].

### 2.7. Cytokine Analysis

Salivary levels of interleukin (IL)-1β, IL-8, monocyte chemoattractant protein-1 (MCP-1), and macrophage migration inhibitory factor (MIF) were measured as previously described [15,16]. In brief, salivary samples were centrifuged at 9300× *g* for 10 min at room temperature. In detection of cytokines, a bead-based immunoassay was used with limits of detection of 0.24 pg/mL for IL-1β, 0.36 pg/mL for IL-8, 2.45 pg/mL for MIF, and 0.44 pg/mL for MCP-1. Altogether, 33 samples of MIF were below the detection limit of 2.45 pg/mL and were, therefore, replaced with limit of detection/2 (LOD/2), while three participants (placebo *n* = 2, probiotics *n* = 1) were completely removed from the analysis as their samples at all three sampling times were below the detection limit.

### 2.8. Protein and Enzyme Analysis

Saliva samples were centrifuged at 4 °C for 10 min at 10,000× *g*, and then three aliquots were made from saliva supernatant for bicinchoninic acid assay (BCA), amylase, total protease activity, chitinase, and albumin to avoid freeze–thaw cycles. The samples were stored at −20 °C until further analysis.

Per the manufacturer’s instructions, the total protein concentration was analyzed as described [17,18] using PierceTM BCA Protein Assay Kit (ThermoFisher, West Palm Beach, FL, USA, Cat#23227). The saliva samples were diluted 1:4 in phosphate-buffered saline (PBS) before testing, and on each plate, a bovine serum albumin (BSA) concentration series was included as a positive control and used to calculate the protein concentration in samples using BSA standard (BSA: 25 μg/mL–1500 μg/mL).

To measure the amylase activity, saliva samples were diluted 1:100 in MILLI-Q before testing. Next, 10 µL of the diluted saliva samples were mixed with 90 µL of a diluted alpha-amylase substrate consisting of 2-CHLORO-4-NITROPHENYL ALPHA-D-MALTOTRIOSIDE (Apollo Scientific, Denton, UK, BITJ00020) in assay buffer: 100 nm MES, 350 nM NaCl, 6 nM Calcium Acetate, 900 nM Potassium Thiocyanate, 0.1% with a final concentration of 1.8 nM Sodium Azide. Absorbance was measured at 450 nm for 15 min with 1 min intervals on a plate reader. For every run, 10 µL of an alpha-amylase standard of 1 U/mL (Sigma Aldrich, Saint Louis, MO, USA, 1 U/mL) mixed with 90 µL of the diluted alpha-amylase substrate was used as a positive control and to calculate U/mL of samples.

Total protease activity (TPA) was measured using PEK-54 substrate ([FITC]-NIeKKKKVLPIQLNAATDK-[KDbc]) with a working solution of 32 μM diluted in TBS. Next, 50 µL of saliva and 50 μL of PEK-54 substrate were added to a black 96-well plate (non-binding), resulting in a final PEK-54 substrate concentration of 16 μM. Reaction mixes were placed in a fluorescence microplate reader with a 485 nm excitation filter and 530 nm emission filter (gain 800), and fluorescence was measured for approximately 1 h at 37 °C using 5 min scanning intervals. Proteolytic activity was expressed as the increase in fluorescence per min (F/min).

To measure the chitinase activity, 50 µL 4-Methylumbelliferyl b-D-N, N′, N″-triacetylchitotriose substrate with a final concentration of 12.7 µM was mixed with 50 µL saliva in a black 96-well plate (non-binding). In every plate, a chitinase control enzyme with a concentration of 0.001 mg/mL was used as a positive control, while a substrate without samples or assay buffer was used as a negative control. Reaction mixes were placed in the fluorescence microplate reader with a 360 nm excitation filter and 450 nm emission filter, and fluorescence was measured for approximately 1 h at 37 °C using 5 min intervals.

Albumin was measured as previously described [17,18] with some modifications. In brief, microplates were coated with rabbit anti-human albumin, cat# A0001 DAKO. Timepoint 1 dilution series of saliva samples were tested (1:500 to 1:6400) to obtain the best sample-dilution factor. Later, dilution 1:2000 was used for all samples. An albumin concentration series was included as a positive control on each plate and used to calculate the albumin concentration in samples (concentration: 0.1 µL/mL–0.00156 µL/mL). Anti-human albumin (horseradish peroxidase (HRP)), (Biorbyt via Bioconnect, Huissen, The Netherlands, Cat# ORB243267) was used as conjugate, and albumin was detected with OPD Substrate tablets (o-phenylenediamine dihydrochloride) (Thermo Fisher, West Palm Beach, FL, USA, #34006) according to protocol.

### 2.9. Bioinformatic Processing and Statistics

Clinical data, plaque score, and BOP % were processed between groups using multiple linear regression (ANCOVA). Changes in plaque and BOP % within groups were compared with analysis of variance (ANOVA) using Tukey correction and visualized in a spaghetti plot. Calculations and visualizations were made in R (version 4.2.3) using R studio (IDE version 2023.04.1 + 446). 

The supragingival microbiota was as previously [8,10,11] characterized and compared according to relative abundance, with data corrected for multiple testing using Benjamini–Hochberg’s correction [19]. Statistically significant differences were tested using Linear discriminant analysis effect Size (LEfSe), whereas compositional changes in the supragingival microbiota were visualized using principal component analysis (PCA). Bioinformatic processing was performed in RStudio IDE (2022.7.1 + 554) running R version 4.1.0. Alpha diversity was compared between groups in R (version 4.3.0) using R studio (IDE version 2023.06.0 + 421) on day 14 and day 28 using linear regression adjusted for baseline values, and the results were visualized for each group. Comparisons within groups were made with ANOVA using Tukey correction. 

Data on cytokines and proteases were log-transformed and compared within groups using repeated-measures one-way ANOVA with Tukey correction, while differences between groups were compared using ANCOVA with baseline measurements as covariates using R (version 4.2.2) and R studio (IDE version 2023.06.2 Build 561). A *p* value < 0.05 was considered significant for all analyses.

## 3. Results

### 3.1. Background Data

Eighty systemically and orally healthy individuals constituted the study population, and all completed the trial. Females comprised the largest proportion of the study population, distributed as 23/40 (58%) in the placebo group and 29/40 (73%) in the probiotic group. The mean age in both the probiotics and placebo groups was 24 years with a range of 19–33 years in the placebo group and 20–30 years in the probiotics group.

### 3.2. Sequencing Metadata

Sample analysis was successful for 238/240 samples (99.2%), yielding between 8154 and 602,392 DNA reads after quality check and bioinformatic processing. From the 238 samples, 13,297,945 million reads were retrieved, and 3084 unique OTUs were identified. A total of 95 different bacterial genera were identified corresponding to 96.21% coverage of the generated sequences, while 333 different bacterial species were identified corresponding to 62.68% coverage of the generated sequences. Alpha diversity, determined using the Shannon and Simpson indices, by mean and range was 4.06 (2.18 to 5.4) and 3.19 (1.15 to 4.47) in the placebo group and 4.08 (1.27 to 5.27) and 3.28 (0.52 to 4.97) in the probiotics group, respectively.

### 3.3. Impact of Oral Hygiene Neglect on Oral Homeostasis in the Placebo Group

As shown in Table 1, 14 days of oral hygiene discontinuation induced significant increases in plaque and BOP % from 0.92 to 2.55 and 5.36% to 24.56%, respectively (plaque *p* < 0.001, BOP % *p* < 0.001). On day 28, plaque levels returned to baseline conditions (0.92 at baseline and 0.88 at day 28, *p* = 0.7), whereas on day 28, BOP % remained significantly elevated compared to the baseline BOP % (5.36% at baseline and 8.77% at day 28, *p* = 0.04).

Microbiologically, 14 days of oral hygiene neglect caused a significant increase in *Capnocytophaga* (*p* < 0.001), *Leptotrichia* (*p* < 0.001), *Selenomonas* (*p* = 0.002), and *Fusobacterium* (*p* = 0.002) genera concomitant with a significant decrease in *Veillonella* (*p* = 0.004)*, Corynebacterium* (*p* = 0.02), *Actinomyces* (0.02), *Streptococcus* (*p* = 0.02), *Rothia* (*p* < 0.001), and *Kingella* (*p* = 0.002) genera. The resume of oral hygiene procedures resulted in a significant increase in the relative abundance of *Streptococcus* (*p* < 0.001), *Veillonella* (*p* < 0.001), *Rothia* (*p* < 0.001), *Corynebacterium* (*p* < 0.001), *Actinomyces* (*p* < 0.001), and *Kingella* (*p* < 0.001) genera together with a significant decrease in the relative abundance of *Capnocytophaga* (*p* < 0.001) and *Leptotrichia* (*p* = 0.03).

The relative abundance of *Streptococcus* (*p* < 0.001), *Rothia* (*p* < 0.009), *Actinomyces* (*p* = 0.002), and *Kingella* (*p* = 0.009) genera remained significantly elevated on day 28 compared to baseline, concomitant with a significantly lower abundance of *Fusobacterium* (*p* = 0.02) genera. Oral hygiene neglect and resumption of oral hygiene procedures had no effect on α-diversity as measured with the Shannon and Simpson indices.

Immunologically, oral hygiene neglect had no significant impact on salivary cytokine levels, which remained stable from baseline to day 28. On the contrary, amylase and albumin decreased significantly during oral hygiene neglect in the placebo group (*p* < 0.001 and *p* = 0.02) and remained significantly lower at day 28 compared to baseline (*p* = 0.04, *p* < 0.001). Chitinase activity was, likewise, significantly lower at day 28 compared to baseline (*p* < 0.001) for the placebo group.

### 3.4. Clinical Effect of Supplementary Consumption of Probiotics

Changes in levels of plaque and BOP % are presented in Table 1 and visualized in Figure 2. Levels of plaque and BOP % increased significantly by day 14 and decreased significantly by day 28 in the probiotics group (*p* < 0.001, *p* < 0.001). Opposite to the placebo group, BOP % did not differ significantly at day 28 compared to baseline in the probiotics group (*p* = 0.2). Although the levels of plaque and BOP % were lower in the probiotics group at all time points, no significant differences between groups were present at any time (plaque, baseline to day 14 *p* = 0.06, baseline to day 28 *p* = 0.57) (BOP % baseline to day 14 *p* = 0.39, baseline to day 28 *p* = 0.16).

### 3.5. Microbiological Effect of Supplementary Consumption of Probiotics

No significant difference in the relative abundance of the predominant genera was observed between groups after 14 days of oral hygiene discontinuation (Figure 3), with the PCA showing a completely random distribution of samples (Figure 4A). LEfSe analysis identified six species which differed significantly between groups on day 14 (Figure 4C).

At day 28, the completely random distribution of the samples from the placebo and the probiotics groups persisted (Figure 4B). Yet, a significantly lower relative abundance of *Lautropia*, *Prevotella*, *Fusobacterium*, and *Selenomonas* genera (*p* = 0.014, *p* = 0.046, *p* = 0.033, *p* = 0.0078, respectively) together with a significantly higher relative abundance of *Rothia* was recorded in the probiotics group at day 28 (*p* = 0.047) (Figure 3, T3). Additionally, seven species were identified with LEfSe as differing significantly between groups on day 28 (Figure 4D).

Comparing α-diversity between the placebo and probiotics groups did not reveal any differences in diversity during the 14 days with oral hygiene neglect according to both the Simpson index (*p* = 0.59) and Shannon index (*p* = 0.79) (Figure 5A,B). Likewise, no significant differences were evident when oral hygiene procedures were resumed in the following fourteen days (Simpson index *p* = 1.00, Shannon index *p* = 0.82) (Figure 5C,D).

### 3.6. Immunological Effect of Supplementary Consumption of Probiotics

Salivary cytokine levels are presented in Table 2. No significant differences were observed between groups at any time point. The salivary activity of the proteases, chitinase, and amylase, the TPA, and the albumin concentration are presented in Table 3. Opposite to the placebo group, albumin concentration remained stable from baseline to day 14 (*p* = 0.26), while TPA (dF/dT) was significantly lower at day 28 than baseline in the probiotics group (*p* = 0.03). Nevertheless, there were no significant differences for any other enzyme activity between groups from either baseline to day 14 or baseline to day 28.

## 4. Discussion

Our main finding is that probiotics may contribute to recovering the baseline composition of the supragingival microbiota in the resolution period after biofilm-induced gingivitis (Figure 3, T3). As expected, biofilm-induced gingivitis impacts the composition of the supragingival microbiota by significantly increasing the relative abundance of *Lautropia*, *Fusobacterium*, and *Selenomonas* genera and significantly decreasing the abundance of *Rothia*. These changes were evident in both the placebo and the probiotics groups (Figure 3, T2). Moreover, this is consistent with previous studies demonstrating that *Prevotella*, *Selenomonas*, and *Fusobacterium* genera are associated with gingivitis, while *Rothia* is associated with oral health [20,21,22]. When comparing the data after 14 days of stopping oral hygiene, no significant differences were observed between the probiotics and the placebo groups (Figure 3, T2) besides the blossoming of potential opportunistic pathogens, such as *Lautropia mirabilis* in the placebo group and *Capnocytophaga gingivalis* and *Capnocytophaga granulosa* [23] in the probiotics group (Figure 4C). Therefore, the data suggest that the consumption of probiotic supplements was unable to prevent the immediate detrimental effect on the supragingival microbiota mediated by biofilm-mediated gingivitis.

It is, therefore, noteworthy that *Lautropia*, *Prevotella*, *Fusobacterium*, and *Selenomonas* genera, which were all significantly increased in the established gingivitis in both groups (day 14), remained significantly higher in the placebo group after 14 days with regular oral care (day 28). In addition, the genus *Rothia*, which was significantly decreased in the gingivitis in both groups (day 14), stayed significantly lower in the placebo group at day 28 (Figure 3, T3). Along this line, *L. mirabilis*, *Rothia auria*, and *Streptococcus sanguinis*, all members of the resident oral microbiota in health [24,25], were all associated with the probiotics group after the resolution period (day 28). On the contrary, *Campylobacter gracilis*, a species associated with periodontitis [26], was associated with the placebo group at day 28 (Figure 4D). Consequently, the data suggest that probiotics augment the conversion to a healthy baseline condition of the supragingival microbiota in the resolution period after biofilm-mediated gingivitis.

As expected, oral hygiene discontinuation had a massive clinical impact (Table 1, Figure 2). However, consumption of probiotic supplements did not have any clinical effect, as evaluated using plaque score and BOP %, which remained comparable between groups on both day 14 and day 28 (Table 1). The only tendency of a clinical difference was at day 14 with borderline less plaque in the probiotics group (*p* = 0.06). In general, this finding is in line with other reports showing little to no impact of probiotics on plaque indices, while a few studies have found significant improvements in BOP % in the probiotics group [3,4,5,6]. Taken together, our findings, therefore, underline that the main effect of probiotics is executed at the microbiological level, i.e., the composition of the oral microbiota, rather than the amount of supragingival plaque.

In the present study, salivary albumin concentration, chitinase and amylase activity, and TPA remained comparable between groups at days 14 and 28 (Table 3), although the level of albumin decreased from baseline to day 14 only in the placebo group, while TPA was significantly lower in the probiotics group at day 28 compared to baseline (Table 3). Salivary albumin, chitinase and amylase activity, and TPA are reported to be higher in individuals with periodontitis as compared to healthy individuals [27,28,29], and our findings are, therefore, surprising, as we observed a significant decrease in the activity of amylase from baseline to day 14 in both groups and albumin concentration in the placebo group. Likewise, the significantly lower levels of albumin, chitinase activity, and amylase activity at day 28 compared to baseline in both groups and TPA in the probiotics group are inconsistent with the perception of a prolonged period with elevated inflammatory activity after inflammatory disease. However, to the best of our knowledge, no studies evaluating probiotics and gingivitis or gingivitis itself have previously investigated the impact on salivary proteases; hence, we cannot compare our results yet.

On the other hand, we were not able to detect any impact of 14 days of oral hygiene neglect on the salivary IL-1β, IL-8, MIF, or MCP-1 levels in either the placebo or the probiotics groups (Table 2), which is contrary to our earlier study where IL-1β, IL-8, and MCP-1 decreased during experimental gingivitis [30]. In contrast, other previous gingivitis studies have shown that patients who return to gingival health after treatment continue to have inflammatory mediators in saliva for weeks after returning to healthy clinical conditions, indicating that longer interventions and follow-ups are needed to detect any changes [31,32]. Few studies have investigated the impact of experimental gingivitis or treatment on IL-1β, IL-8, or MCP-1 levels [33,34,35,36,37]. Notably, in previous studies, measurements were based on gingival crevicular fluid (GCF) samples, which hampers comparison with our saliva-based data. Likewise, only a few studies investigated the impact of probiotics and gingivitis on cytokine levels; some found a favoring of the probiotics group [33,34,35], while others found no significant difference between the probiotics and placebo groups [36,37]. Our results support the latter finding, showing that supplementing with probiotics does not lead to significant increases or decreases in IL-1β, IL-8, or MCP-1 levels in individuals without clinical attachment loss. This finding could be an expression that the suppressive effect of probiotics observed on the oral microbiota was not an indirect effect through an upregulated immune response but rather a direct impact on the oral microbiota. Nevertheless, we were not able to fully evaluate the effect of the tested probiotics on the host response as intended.

Accordingly, several limitations apply to the present study including the monitoring of solely pro-inflammatory cytokines, of which we were unable to detect any impact. Therefore, future probiotics studies should consider monitoring different pro-inflammatory cytokines such as TNF-α and anti-inflammatory cytokines including IL-10 and IL-17 and should use GCF instead of saliva samples. However, if evaluating GCF, it is essential to note that the evaluation would be site-specific. Another limitation was the size of the cohort which, despite a considerable sample size, turned out to be underpowered in terms of the clinical endpoints tested. Specifically, a post hoc analysis of the clinical data performed with twice the sample size showed a significant difference between groups in terms of the plaque score at day 14 (*p* = 0.009) and the BOP % at day 28 (*p* = 0.04)—both favoring the probiotics group. Importantly, the power calculation of the present study was based on microbiological data [9]. Finally, our results from a young and healthy study population cannot be transferred to an ill or old population with a weakened immune response. However, we, for ethical reasons, only found it possible to perform the study in a young and healthy population. Thus, future studies with larger sample sizes are needed to evaluate the clinical effect of the tested probiotic supplement. 

## 5. Conclusions

In conclusion, the data from the present study confirm the hypothesis that probiotics support recovery toward a baseline composition of the supragingival microbiota during the resolution of biofilm-mediated gingivitis. Future studies with larger sample sizes are needed to evaluate the clinical significance of these findings.

## Figures and Tables

**Figure 1 nutrients-15-04805-f001:**
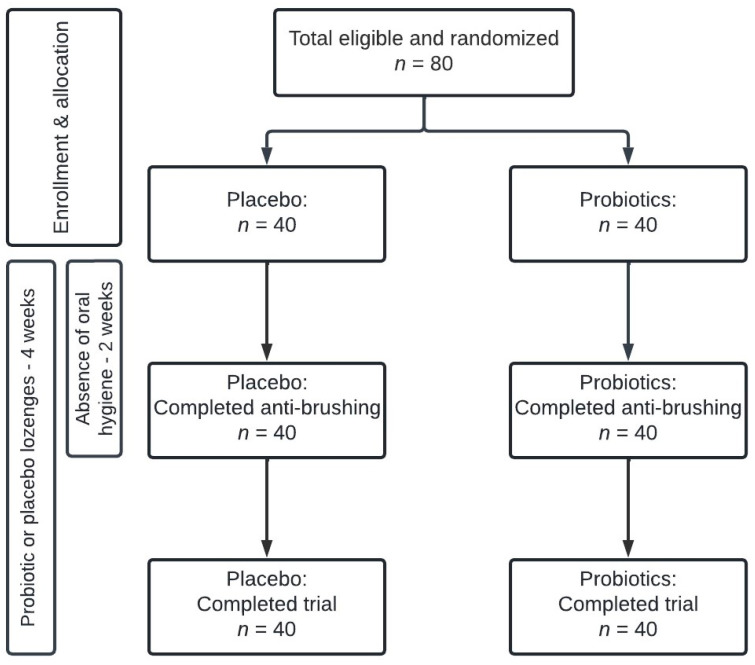
Flowchart of the study.

**Figure 2 nutrients-15-04805-f002:**
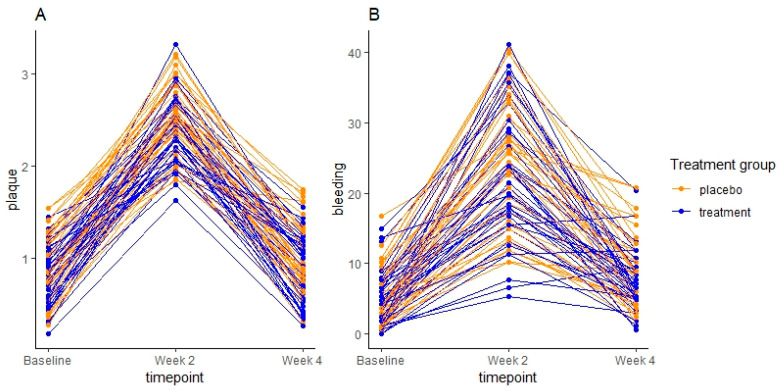
Spaghetti plot of levels of plaque (**A**) and bleeding on probing in percentage (**B**) measured at baseline, day 4 (week 2), and day 28 (week 4) for placebo (orange) and probiotics (blue).

**Figure 3 nutrients-15-04805-f003:**
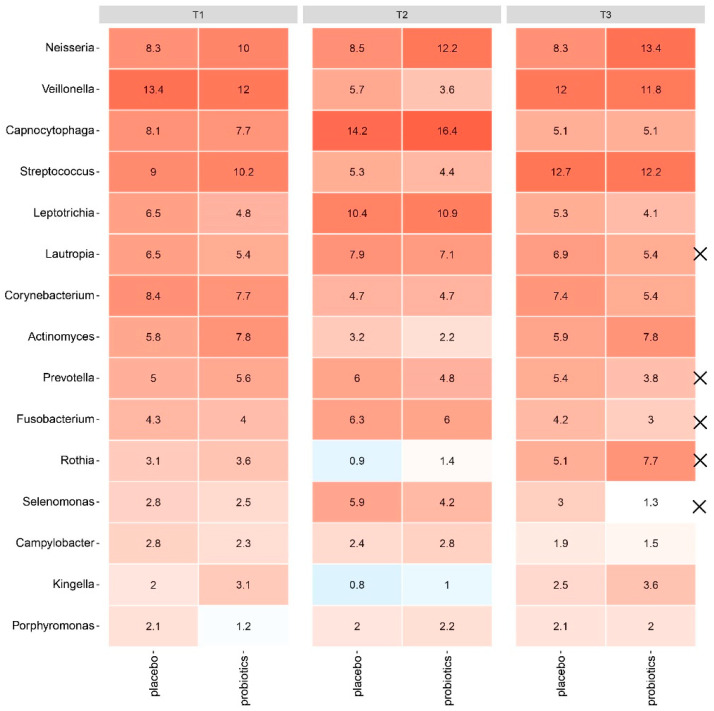
Impact of sugar stress on the 15 most abundant genera in the supragingival microbiota for the placebo and probiotics groups. X marks significant differences between groups. Intensity of coloration marks increasing abundance.

**Figure 4 nutrients-15-04805-f004:**
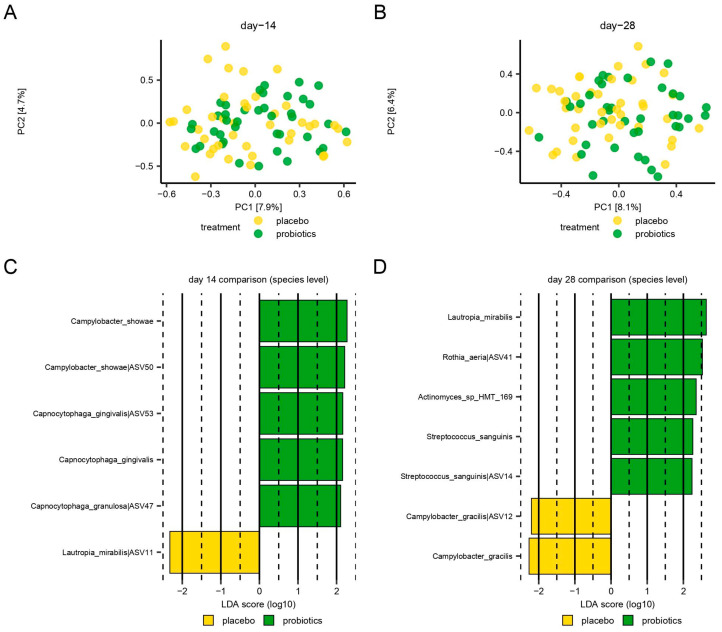
Compositional changes incited by the absence of oral hygiene. Principal component analysis (PCA) was expressed using the two most decisive components (PC1 and PC2), which covered approximately 8% of the variation of the dataset. Placebo vs. probiotics day 14 (**A**). Placebo vs. probiotics day 28 (**B**). Linear discriminant analysis effect size (LEfSe) analysis expressed using significant species at day 14 (**C**) and day 28 (**D**).

**Figure 5 nutrients-15-04805-f005:**
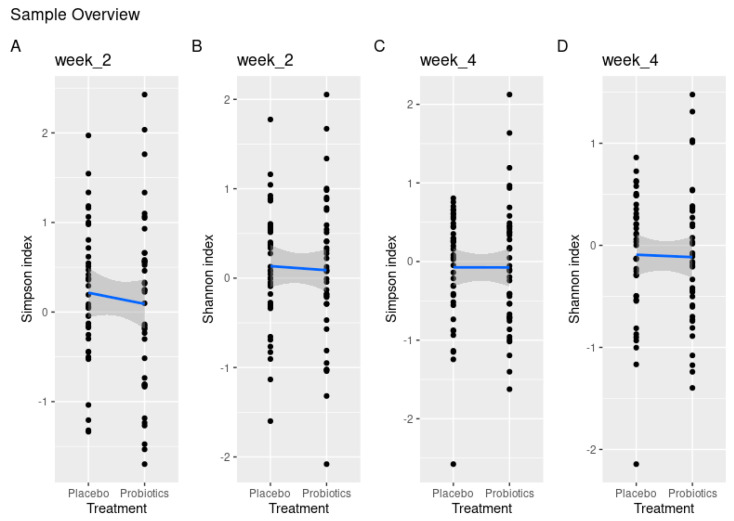
Treatment effect measured according to disparities in alpha diversity demonstrated by the Shannon and Simpson index values between placebo and probiotics after 14 days of oral hygiene neglect (**A**,**B**) and 14 days of resuming oral hygiene at day 28 (**C**,**D**).

**Table 1 nutrients-15-04805-t001:** Background information of the study population. Clinical data measured according to levels of plaque and BOP %. * Significant differences.

	Placebo (*n* = 40)	Probiotic (*n* = 40)	*p* Value [CI] between Groups
Sex (female/male)	29/11	23/17	
Age	24 (19–33)	24 (20–30)	
Clinical examination			
Baseline			
Plaque mean (sd)	0.92 (0.3)	0.8 (0.3)	
BOP % mean (sd)	5.36 (3.9)	4.73 (3.7)	
Day 14			
Plaque mean (sd)	2.55 (0.4)	2.37 (0.3)	0.06 [−0.3; 0.01]
Baseline vs. day 14 *p* value [CI]	<0.001 * [1.4; 1.8]	<0.001 * [1.4; 1.8]	
BOP % mean (sd)	24.56 (8.6)	22.34 (9.2)	0.39 [−5.2; 2]
Baseline vs. day 14 *p* value [CI]	<0.001 * [15.9; 22.5]	<0.001 * [14.2; 20.9]	
Day 28			
Plaque mean (sd)	0.88 (0.4)	0.86 (0.4)	0.57 [−0.2; 0.1]
Day 14 vs. day 28 *p* value [CI]	<0.001 * [−1.8; −1.4]	<0.001 * [−1.7; −1.3]	
Baseline vs. day 28 *p* value [CI]	0.7 [−0.1; 0.3]	0.7 [−0.1; 0.2]	
BOP % mean (sd)	8.77 (5.1)	7.1 (4.1)	0.16 [−3.3; 0.5]
Day 14 vs. day 28 *p* value [CI]	<0.001 * [−19.1; −12.5]	<0.001 * [−18.5; −12]	
Baseline vs. day 28 *p* value [CI]	0.04 * [0.1; 6.7]	0.2 [−0.9; 5.6]	

**Table 2 nutrients-15-04805-t002:** Salivary levels of IL-1β, IL-8, MIF, and MCP-1 measured in pg/mL. Data is log10 transformed and presented as means and standard deviation (sd) for the probiotics and placebo groups on day 14 and day 28; *p* values and 95% confidence intervals [CI] are presented for comparison within groups from baseline vs. day 14 and baseline vs. day 28 performed using ANOVA with baseline values as covariates; *p* values and 95% confidence intervals [CI] for comparison by group were performed using ANCOVA with baseline values as covariates.

	IL-1β	IL-8	MCP-1	MIF
Baseline				
Placebo mean (sd)	1.14 (0.51)	2.25 (0.43)	1.5 (0.42)	1.6 (0.81)
Probiotics mean (sd)	1.08 (0.45)	2.15 (0.38)	1.38 (0.41)	1.6 (0.77)
Day 14				
Placebo mean (sd)	1.16 (0.45)	2.23 (0.40)	1.60 (0.42)	1.74 (0.79)
Baseline vs. day 14 *p* value [CI]	0.97 [−0.5; 0.18]	0.95 [−0.17; 0.13]	0.31 [−0.06; 0.26]	0.57 [−0.18; 0.46]
Probiotics mean (sd)	1.09 (0.45)	2.09 (0.40)	1.47 (0.40)	1.49 (0.95)
Baseline vs. day 14 *p* value [CI]	0.99 [−0.13; 0.15]	0.44 [−0.19; 0.06]	0.17 [−0.03; 0.22]	0.67 [−0.43; 0.2]
Between groups *p* value [CI]	0.67 [−0.19; 0.12]	0.24 [−0.22; 0.06]	0.47 [−0.21; 0.10]	0.15 [−0.60; 0.10]
Day 28				
Placebo mean (sd)	1.12 (0.54)	2.21 (0.46)	1.45 (0.40)	1.68 (0.82)
Baseline vs. day 28 *p* value [CI]	0.93 [−0.19; 0.14]	0.77 [−0.19; 0.11]	0.76 [−0.21; 0.11]	0.82 [−0.24; 0.4]
Probiotics mean (sd)	1.03 (0.48)	2.12 (0.48)	1.42 (0.44)	1.58 (0.95)
Baseline vs. day 28 *p* value [CI]	0.65 [−0.19; 0.09]	0.82 [−0.15; 0.09]	0.65 [−0.08; 0.17]	0.98 [−0.34; 0.29]
Between groups *p* value [CI]	0.60 [−0.22; 0.13]	0.84 [−0.18; 0.15]	0.52 [−0.10; 0.20]	0.52 [−0.42; 0.21]

**Table 3 nutrients-15-04805-t003:** Salivary levels of amylase activity, chitinase activity, total protease activity (TPA), and albumin concentration measured in slope/min, dF/dT, slope/sec, and µg/mL, respectively. Data is log10 transformed and presented as means and standard deviation (sd) for the probiotics and placebo groups on day 14 and day 28; *p* values and 95% confidence intervals [CI] are presented for comparison within groups from baseline vs. day 14 and baseline vs. day 28 performed using ANOVA with baseline values as covariates; *p* values and 95% confidence intervals (CI) are presented for comparison by group performed using ANCOVA with baseline values as covariates. * Significant differences.

	Amylase Activity (Slope/Min)	Total Protease Activity (dF/dT)	Chitinase Activity (Slope/s)	Albumin (µg/mL)
Baseline				
Placebo mean (sd)	0.04 (0.03)	3.12 (0.32)	0.29 (0.19)	1.82 (0.31)
Probiotics mean (sd)	0.05 (0.02)	3.1 (0.22)	0.34 (0.03)	1.77 (0.27)
Day 14				
Placebo mean (sd)	0.03 (0.01)	3.10 (0.24)	0.26 (0.17)	1.69 (0.25)
Baseline vs. day 14 *p* value [CI]	<0.001 * [−0.03; 0.0006]	0.84 [−0.14; 0.08]	0.33 [−0.07; 0.02]	0.02 * [−0.24; −0.02]
Probiotics mean (sd)	0.04 (0.02)	3.09 (0.19)	0.31 (0.19)	1.70 (0.29)
Baseline vs. day 14 *p* value [CI]	<0.001 * [−0.02; −0.0005]	0.98 [−0.1; 0.08]	0.48 [−0.07; 0.02]	0.26 [−0.17; 0.03]
Between groups *p* value [CI]	0.20 [−0.003; 0.01]	0.99 [−0.09; 0.09]	0.48 [−0.03; 0.07]	0.59 [−0.08; 0.15]
Day 28				
Placebo mean (sd)	0.04 (0.02)	3.05 (0.23)	0.22 (0.16)	1.60 (0.32)
Baseline vs. day 28 *p* value [CI]	0.04 * [−0.03; 0.0004]	0.25 [−0.18; 0.04]	<0.001 * [−0.11; −0.02]	<0.001 * [0.33; −0.11]
Probiotics mean (sd)	0.04 (0.02)	3.00 (0.23)	0.25 (0.15)	1.61 (0.26)
Baseline vs. day 14 *p* value [CI]	0.009 * [−0.02; −0.002]	0.03 * [−0.19; −0.007]	<0.001 * [−0.14; −0.04]	0.001 * [−0.26; −0,05]
Between groups *p* value [CI]	0.54 [−0.005; 0.009]	0.41 [−0.14; 0.06]	0.96 [−0.04; 0.04]	0.38 [−0.06; 0.15]

## Data Availability

Raw sequences have been deposited in the European Nucleotide Archive (ENA, www.ebi.ac.uk, accessed on 8 November 2023) with the accession number PRJEB69273. Raw cytokine, protease, and albumin data along with compositional information about the probiotic and placebo lozenges are available upon request from the corresponding author. Likewise, the full trial protocol is available in Danish upon request from the corresponding author.

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
