# Peer review of "Probiotics Support Resilience of the Oral Microbiota during Resolution after Experimental Gingivitis—A Randomized, Double-Blinded, Placebo-Controlled Trial"

_nutrients, 2023, doi:10.3390/nu15224805_

Round 1
Reviewer 1 Report
Comments and Suggestions for Authors
I was pleased to review the paper entitled Probiotics support resilience of the oral microbiota during resolution after experimental gingivitis – a randomized, double-blinded, placebo-controlled trial
This manuscript is highly interesting and novel and touches on a highly important question. I find the experiment well organized and manuscript clearly written. I am strongly inclined to recommend this article for publication in Nutrients.
I only have minor questions and suggestions to the authors:
Abstract
1. Lines 18-19: “gingivitis-induced oral dysbiosis”
This phrase sounds a bit confusing. What is the cause and what is the consequence: gingivitis or dysbiosis?
2. Line 28: text formatting is required.
3. Keywords: It is recommended to use MeSH terms and alphabetic order.
Materials and methods
4. Table 1: are the differences between the groups and visits significant? Please add p-values to the table, not only to the text.
Discussion
5. I suggest that the authors should extend the Discussion section. Please provide more details about the results of other studies assessing the impact of probiotics on plaque indices and gingival condition (BOP), as these results are controversial.
Author Response
Reviewer 1:
Abstract
- Lines 18-19: “gingivitis-induced oral dysbiosis” This phrase sounds a bit confusing. What is the cause and what is the consequence: gingivitis or dysbiosis?
Answer: The line has been updated. “The present study aimed to test if probiotics protect against experimental gingivitis incited by 14 days of oral hygiene neglect, and/or support restoring oral homeostasis subsequently.”
- Line 28: text formatting is required.
Answer: The formatting error has been fixed.
- Keywords: It is recommended to use MeSH terms and alphabetic order.
Answer: The keywords have been updated as requested.
Materials and methods
- Table 1: are the differences between the groups and visits significant? Please add p-values to the table, not only to the text.
Answer: Table 1 has been extended with p-values and confidence intervals between visits and groups.
Discussion
- I suggest that the authors should extend the Discussion section. Please provide more details about the results of other studies assessing the impact of probiotics on plaque indices and gingival condition (BOP), as these results are controversial.
Answer: We thank the reviewer for the suggestion. The details of the results from previous studies have been extended in lines 379-381. However, as the number of studies – especially assessing the same endpoints as us – is still low the amount of details is limited.
“In general, this finding is in line with other reports showing none to minimal impact of probiotics on plaque indexes, while a few studies find significant improvements of BOP % in the probiotic groups”.
Reviewer 2 Report
Comments and Suggestions for Authors
Dear authors!
Indeed, the issue of maintaining normal microflora in inflammatory diseases of the oral cavity is relevant, as evidenced by many researches aimed at studying existing methods of prevention and treatment of these conditions, as well as the development of new ones. The presented study aimed at evaluating the effectiveness of the probiotics use in the initiation and resolution of biofilm-induced gingivitis, is interesting as it considers not only the clinical outcome, but also microbiological and immunological parameters. The obtained results are also of practical value, as they allow to determine the need for the use of probiotics in gingivitis and in the resolution period after gingivitis. In the future, it would be interesting to consider the use of probiotics not only for gingivitis, but also for other inflammatory periodontal diseases, and, possibly, to make clinical recommendations for the use of probiotics.
The "Materials and methods" section is described in sufficient detail to ensure the reproducibility of the research and complies with the necessary requirements, including ethical ones. The design of the study is simple and well understood, and also corresponds to the purpose of the study. The manuscript has a clear structure and is easy to read. The results of the study are presented in detail both in the text and in tables. The conclusions are formulated correctly and correspond to the aim of the study. Literary references are relevant and most of them have been published no earlier than 10 years. Excessive number of self-citations is excluded.
In general, the article leaves a good impression, however there are some questions and recommendations:
1. [Line 28] Pay attention to the design of the article.
2. [Lines 92-93] On the website ClinicalTrials.gov you indicate that patients aged 18-30 years are the study inclusion criteria, which does not correspond to the data presented in the article. Why do not these data converge? Perhaps there is a mistake in the article?
3. Also, you write that this study is double-blinded however on ClinicalTrials.gov in the “Masking” section, you indicate that the study is a quadruple (Participant, Care Provider, Investigator, Outcomes Assessor). Please explain this aspect.
Thus, I can recommend this article for publication after minor edits.
Author Response
Reviewer 2:
- [Line 28] Pay attention to the design of the article.
Answer: The formatting error has been fixed.
- [Lines 92-93] On the website ClinicalTrials.gov you indicate that patients aged 18-30 years are the study inclusion criteria, which does not correspond to the data presented in the article. Why do not these data converge? Perhaps there is a mistake in the article?
Answer: Originally, we aimed to include those from 18-30 years of age. However, during recruitment healthy volunteers aged 31-35 (n=x) offered to participate, which we accepted as they fulfilled all other inclusion criteria. By mistake, this was updated this was not updated on ClinicalTrials.gov. Naturally, we will update the ClinicalTrials.gov record accordingly.
- Also, you write that this study is double-blinded however on ClinicalTrials.gov in the “Masking” section, you indicate that the study is a quadruple (Participant, Care Provider, Investigator, Outcomes Assessor). Please explain this aspect.
Answer: We thank the reviewer for this comment, which is correct, as masking was performed with a quadruple design. However, as the term double-blinded is normally used in study titles, and because most clinicians are familiar with this term, we preferred to use double-blinded in the title. According to you suggestion we have updated the methods section lines 78-79. The present study is a quadruple-blinded (Participant, Care Provider, Investigator, Outcomes Assessor), randomized, placebo-controlled trial.
Reviewer 3 Report
Comments and Suggestions for Authors
Review: PLB.
Journal
Nutrients (ISSN 2072-6643)
Manuscript ID
nutrients-2696849
Type
Article
Title
Probiotics support resilience of the oral microbiota during resolution after experimental gingivitis – a randomized, double-blinded, placebo-controlled trial
Authors
Christine Lundtorp-Olsen * , Laura Massarenti , Vincent Frederik Dahl Vendius , Ulvi Kahraman Gürsoy , Annina Van Splunter , Floris James Bikker , Mervi Gürsoy , Christian Damgaard , Merete Markvart , Daniel Belstrøm
The objective of this research is to measure the effectiveness of probiotics against oral dysbiosis induced by gingivitis following 14 days of neglect in oral hygiene, and/or maintain oral microbial homeostasis. In conclusion, the study puts forward the hypothesis that probiotics maintain the resilience of the oral microbiota.
Why prefer xylitol to chlorhexidine which has nevertheless proven its superiority in terms of oral hygiene?
Several clarifications should be made in the text.
Abstract: not species but genus . In the summary we must find details on the probiotics used.
Line 99: limiting the age to 35 years requires specifying that the immune defenses of these patients are sufficient to limit a harmful reaction. This is not the case for elderly patients whose immune defenses are inherently reduced.
Line 100: if you tolerate including patients with periodontitis, extensive gingivitis (>20%), can this distort your approach. Is the second quadrant involved in this descrimination? Wasn't it simpler to exclude these patients or to perform oral cavity sanitation?
Line 120: more details on the hygienic procedures are necessary (were they identical for all patients?) (what did this procedure consist of, toothpaste, mouthwash, interdental floss, etc., frequencies, duration of brushing, etc.). The composition of toothpastes and mouthwashes can have an influence on fluctuations in the oral microbiota.
Line 128: what is the role of the effectiveness of probiotic and xylitol on the oral microbiota?
Line 149: can translate LOD/2 into full letters for better understanding.
Line 155: can translate BCA in full for better understanding.
Line 245 the mention of bacteria here is at the scale of the genus and not the species.
Line 346: not species but genus !
To enrich the discussion some references can be added.
Krupa NC, Thippeswamy HM, Chandrashekar BR. Antimicrobial efficacy of Xylitol, Probiotic and Chlorhexidine mouth rinses among children and elderly population at high risk for dental caries - A Randomized Controlled Trial. J Prev Med Hyg. 2022 Jul 31;63(2):E282-E287. doi: 10.15167/2421-4248/jpmh2022.63.2.1772. PMID: 35968060; PMCID: PMC9351416.
Gupta, H.; Kim, S.H.; Kim, S.K.; Han, S.H.; Kwon, H.C.; Suk, K.T. Beneficial Shifts in Gut Microbiota by Lacticaseibacillus rhamnosus R0011 and Lactobacillus helveticus R0052 in Alcoholic Hepatitis. Microorganisms 2022, 10, 1474. https://doi.org/ 10.3390/microorganisms10071474
Author Response
Reviewer 3:
- Why prefer xylitol to chlorhexidine which has nevertheless proven its superiority in terms of oral hygiene?
Answer: We thank the reviewer for the comment. In our study, the addition of xylitol is based on our previous study (reference). In the study, we tested the impact of sugar rinses 6-8 times a day for 14 days with concomitant consumption of probiotics (same species as used in this trial) or placebo lozenges twice a day. Furthermore, we had a group of participants who rinsed with xylitol 6-8 times a day with concomitant consumption of probiotics (same species as used in this trial) or placebo lozenges twice a day. Surprisingly, we found that consumption of probiotic lozenges combined with xylitol rinses was able to induce statistically significant health-associated compositional changes to the salivary microbiota in orally healthy individuals, which was antagonistic to sugar-mediated dysbiosis. This information has likewise been added to lines 58-61.
Afterward, it was technically easy to add xylitol as a powder to the lozenges. On the contrary, the addition of chlorhexidine as a liquid is much more complicated. However, based on the recommendations from reviewer 3, we will strongly consider including chlorhexidine in combination with probiotics in future studies.
“Recently, we demonstrated that consumption of probiotic lozenges containing Lacticaseibacillus rhamnosus PB01 DSM14870 and Lactilactobacillus curvatus EB10 DSM32307 combined with xylitol was able to induce statistically significant health-associated compositional changes to the salivary microbiota in orally healthy individuals, which was antagonistic to sugar-mediated dysbiosis”.
- Abstract: not species but genus. In the summary we must find details on the probiotics used.
Answer: We thank the reviewer for the comment. Species have been changed to genus. We respectfully point to the reviewer that the details on the species used in the probiotic lozenges are written in lines 65-66 and lines 129-131. However, we have likewise added in the patens section, that compositional information is available upon request from the corresponding author in lines 457-458.
Line 65-66: “The present study, therefore, tested whether the combination of L.rhamnosus PB01 DSM14870 and L.curvatus EB10 DSM32307 combined with xylitol protects the oral cavity…”
Line 129-131: “In brief, the probiotic lozenges contained an equal mix of L. rhamnosus PB01 DSM14870 and L. curvatus EB10 DSM32307 with a concentration of 1ꞏ109 CFU/tablet and 491 mg xylitol.”
Line 457-458: “Raw cytokine,protease and albumin data along with compositional information about the probiotc and placebo lozenges are available upon request from the corresponding author.”
- Line 99: limiting the age to 35 years requires specifying that the immune defenses of these patients are sufficient to limit a harmful reaction. This is not the case for elderly patients whose immune defenses are inherently reduced.
Answer: We thank the reviewer for the comment. We agree that this is a limitation of the study. However, we did for ethical reasons only find it ethically acceptable to include young and healthy individuals. We have likewise added this information to the limitation section in the discussion lines 430-432.
“Finally, our results from a young and healthy study population cannot be transferred to an ill or old population with a weakened immune response. However, we did for ethical reasons only find it possible to perform the study in a young and healthy population.”
- Line 100: if you tolerate including patients with periodontitis, extensive gingivitis (>20%), can this distort your approach. Is the second quadrant involved in this descrimination? Wasn't it simpler to exclude these patients or to perform oral cavity sanitation?
Answer: We respectfully point to the reviewer that these patients were excluded and not included.
“The inclusion criteria were systemically and orally healthy adults aged between 18-35 years old. Exclusion criteria were active dental caries or periodontitis, extensive gingivitis (>20 %), pregnancy, systemic diseases, use of medication, and systemic antibiotics within the last three months.”
- Line 120: more details on the hygienic procedures are necessary (were they identical for all patients?) (what did this procedure consist of, toothpaste, mouthwash, interdental floss, etc., frequencies, duration of brushing, etc.). The composition of toothpastes and mouthwashes can have an influence on fluctuations in the oral microbiota.
Answer: We thank the reviewer for this comment. All participants received the same instructions, including brushing as well as interdental cleaning twice a day. This information has now been added.
“At baseline and day 28, participants performed their normal oral hygiene procedure (brushing and interdental cleaning twice a day) after the collection of supragingival plaque samples and before clinical examinations”
- Line 128: what is the role of the effectiveness of probiotic and xylitol on the oral microbiota?
Answer: We respectfully point to the reviewer that this matter is presented in the introduction section lines 58-61. However, based on the comment from the referee, we have added additional information about the compositional changes presented in reference 8.
“Recently, we demonstrated that consumption of probiotic lozenges containing Lacticaseibacillus rhamnosus PB01 DSM14870 and Lactilactobacillus curvatus EB10 DSM32307 combined with xylitol was able to induce statistically significant health-associated compositional changes to the salivary microbiota in orally healthy individuals, which was antagonistic to sugar-mediated dysbiosis.”
- Line 149: can translate LOD/2 into full letters for better understanding.
Answer: LOD/2 has been written out in text.
“Altogether 33 samples of MIF were below the detection limit of 2.45 pg/mL and therefore replaced with limit of detection/2 (LOD/2)”
- Line 155: can translate BCA in full for better understanding.
Answer: BCA has been written out in the text.
“three aliquots were made from saliva supernatant for bicinchoninic acid assay (BCA)”
- Line 245 the mention of bacteria here is at the scale of the genus and not the species.
Answer: Species have been changed to genera in general in the result section.
- Line 346: not species but genus !
Answer: Species have been changed to genera in general in the discussion.
- To enrich the discussion some references can be added.
Answer: We thank the reviewer for suggesting very interesting references. However, we chose not to use references on cariogenic end-points, children, and elderly as well as studies assessing other microbiotas than the oral, wherefore we did not add the suggested references.